# Progressive Checkerboards for Autoregressive Multiscale Image Generation

**David Eigen**                                                                              *de@deigen.net*

**Reviewed on OpenReview:** *https://openreview.net/forum?id=wkCTCXo1a9*

## Abstract

A key challenge in autoregressive image generation is to efficiently sample independent locations in parallel, while still modeling mutual dependencies with serial conditioning. Some recent works have addressed this by conditioning between scales in a multiscale pyramid. Others have looked at parallelizing samples in a single image using regular partitions or randomized orders. In this work we examine a flexible, fixed ordering based on *progressive checkerboards* for multiscale autoregressive image generation. Our ordering draws samples in parallel from evenly spaced regions at each scale, maintaining full balance in all levels of a quadtree subdivision at each step. This enables effective conditioning both between and within scales. Intriguingly, we find evidence that in our balanced setting, a wide range of scale-up factors lead to similar results, so long as the total number of serial steps is constant. On class-conditional ImageNet, our method achieves competitive performance compared to recent state-of-the-art autoregressive systems with like model capacity, using fewer sampling steps. Code is available at `https://github.com/deigen/checkerboardgen`.

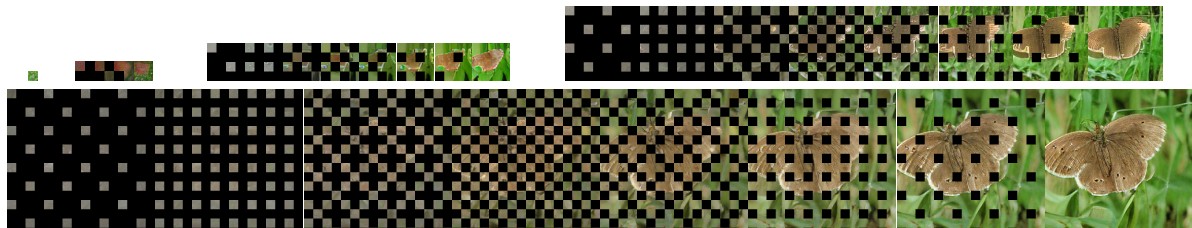

Figure 1: Progressive checkerboard samples from our model using 2x scale factor and 8 steps per scale. Masking applied to sampled locations at each step after decoding for visualization.

## 1 Introduction

Image generation methods, including autoregressive (AR) models, diffusion and masked models, have shown impressive results in generating high-quality images (Ramesh et al., 2021; 2022; Saharia et al., 2022; van den Oord et al., 2016; Salimans et al., 2017; Razavi et al., 2019; Ho et al., 2020; Rombach et al., 2022; Chang et al., 2023; Yu et al., 2024). AR models generate images by sampling a sequence of image tokens from a learned distribution, where each token is conditioned on previously sampled tokens. Recent AR methods include scalewise autoregressive models (Tian et al., 2024; Han et al., 2025; Ma et al., 2025), which condition from coarse to fine as their progressive sequence, as well as parallel AR models (Wang et al., 2025; Zhang et al., 2025), which select multiple locations at once by conditioning with sequential blocks. Both conditioning methods aim to model dependencies between tokens in a way that allows for sampling multiple locations in parallel, which is critical for fast generation.

However, samples drawn in parallel are also drawn independently from one another, which can produce inconsistencies when they are mutually dependent. This is particularly true for adjacent and nearby locations. For example, if an image patch is red, the one next to it is likely to be red as well, but drawing tokens

independently may choose two different colors instead of the same one. Conditioning each location on a common parent, for example in the previous scale of a scalewise progression, makes the conditional samples more independent of each other (Tian et al., 2024).

Relying exclusively on scalewise conditioning, however, depends on a slow scale-up factor: If the scale factor is too large, then an object spanning multiple locations in one scale may not yet be visible in the previous scale, so its appearance dependencies would not be conditioned and independently sampling can mix modes. Rather than scaling up by a factor of 2, the best performing multiscale models (Tian et al., 2024; Han et al., 2025; Ma et al., 2025) currently scale by a factor of $\sqrt[3]{2} \approx 1.26$ (i.e. each factor of 2 is further subdivided into three scales). However, as prior sequential and parallel methods (Wang et al., 2025; Zhang et al., 2025) have shown, another way to condition is to model dependencies between sampling locations.

In this paper, we examine conditioning between locations *within* each scale, making use of both between- and within-scale conditioning. We develop a sampling order based on a progressive checkerboard, which reduces mutual dependence within each sampling block and enables fast scale-up. Our ordering maintains balance at all levels in a quadtree subdivision, so that varying the block size effectively varies the trade-off between parallelism and modeling conditional dependencies. We use this to explore the relationships between scale-wise and within-scale conditioning, finding somewhat surprisingly that for this spatially balanced setup, the total number of sequential steps largely determines performance independently of how the steps are divided among scales. Our model is competitive with state-of-the-art methods in ImageNet 256x256 class-conditioned generation, using just 17 sampling steps.

## 2 Background and Related Work

### 2.1 Background: Autoregression for Images

Autoregressive (AR) models for images sample a sequence of tokens from a learned distribution, where each token is conditioned on previously sampled ones. Tokens representing images are typically formed using a VAE autoencoder (Razavi et al., 2019; Rombach et al., 2022), which maps between RGB images and local patch representations that are clustered into a discrete set of codes. The AR model then learns distributions over token codes $z_t$. In the simplest case, this is done one at a time in a conditional sequence, $P(z_1, z_2, \ldots, z_N) = P(z_1)P(z_2|z_1)\ldots P(z_N|z_1, z_2, \ldots, z_{N-1})$. Typically, the distribution is modeled using a transformer (Vaswani et al., 2017; Radford et al., 2018; 2019). To generate an image, we draw a sequence of $N$ tokens from the model, where each token $t$ is sampled from $P(z_t|z_1, \ldots, z_{t-1})$. The VAE decoder converts the token representations into an RGB image. However, sampling tokens one at a time can be slow. Thus, recent methods have explored ways to speed up sampling by computing multiple tokens in parallel.

### 2.2 Scalewise Autoregression

Gradual scaling AR models (Tian et al., 2024; Ma et al., 2025; Han et al., 2025), introduced by VAR (Tian et al., 2024), progressively develop a full-resolution array of latent codes, which is used as a working "canvas" to apply residuals. At each iteration, the canvas is downsampled to the current scale (which grows by a factor of $\sqrt[3]{2}$) and a transformer predicts residuals, which are upsampled and applied to the full-resolution array.

The downsample-predict-upsample cycle implicitly invokes a progressive deblurring in latent space, similar to the progressive denoising of diffusion models. Viewing the method this way sheds some light on reasons why the scaling factor remains small. Just as diffusion models struggle with large denoising steps due to their use of independent sampling and Gaussian noise (Song et al., 2021; Salimans & Ho, 2022; Song et al., 2023; Lu et al., 2022), scalewise AR models can struggle with large deblurring steps due to independently sampling underlying multi-modal distributions (Bansal et al., 2023; Hoogeboom & Salimans, 2022).

### 2.3 Parallel Sampling

Parallel autoregression (PAR) by Wang et al. (2025) generates images by sampling in parallel from equal-sized square partitions. Their method uses raster order within each partition and no multi-scale sampling, limiting its parallelism to only four groups. More recently, Zhang et al. (2025) use a locality-aware ordering that adds far-away locations while growing already-selected regions with adjacent samples, while Pang et al. (2025) use random orderings. However, the former uses a more complex dynamic evolution to grow regions, while both

methods require additional position probe tokens, extending the overall sequence length. We use a simple but effective regular pattern with multiscale conditioning, and do not require probe tokens.

Yan et al. (2025) use a two-stage "generation then reconstruction" approach applied in the context of MAR (Li et al., 2024) that applies unmasking using diagonal striations, the final stage of which corresponds to a mod 2 checkerboard. In contrast to their method, we use a fully balanced progressive checkerboard, along with explicit multiscale conditioning, in the context of direct autoregression instead of masking. Additional recent works include xAR (Ren et al., 2025b), which uses flow-matching to generate large chunks at each AR step; NAR (He et al., 2025), which uses horizontal and vertical decoders to condition in striations; and ARPG (Li et al., 2025), which uses cross-attention to condition random-order samples with masked models.

Some gradual scaling models also incorporate ways to reduce same-scale dependencies. Han et al. (2025) use random bit flips to address errors from independent sampling at the quantizer bit level after incorporating BSQ (Zhao et al., 2025) as their tokenizer. Ma et al. (2025) use a miniature masked image model applied in the autoregressor head to condition lower-confidence locations on higher-confidence ones. To enable 2x scaling, Ren et al. (2025a) stack a flow-matching model over mixed-mode AR samples to resolve interdependences. Kutscher et al. (2025) examine the impact of patch ordering in the context of recognition models.

## 3 Method

### 3.1 Autoregressor

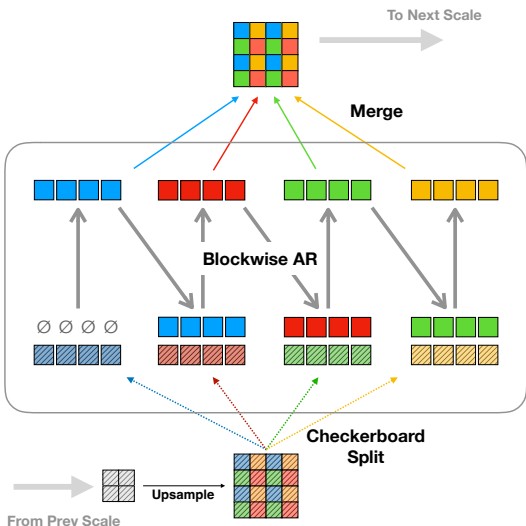

Figure 2: Overview of our multiscale blockwise checkerboard autoregressor.

Our multiscale progressive checkerboard autoregressive model is built around a transformer (Vaswani et al., 2017; Radford et al., 2018; 2019) with blockwise causal mask using successive checkerboard sampling blocks.

Figure 2 provides an overview of our method. At each scale $s$, latent codes output by the previous scale $s-1$ are upsampled by a scaling factor $r$ to form the scalewise-conditioning input, $z_{s-1}^{up} = \text{upsample}(z_{s-1}, r)$. The locations of this map are split according to the progressive checkerboard ordering (see Sec. 3.2) into $P$ blocks, $b_s^1, b_s^2, \ldots, b_s^P$, where each block $b_s^i$ contains $H_s W_s / P$ tokens. We randomize $P$ during training to allow different degrees of parallelism at inference time.

In order for each autoregressive step to condition on previously sampled tokens *within* each scale, we include the output of each block into the input of later ones. In particular, we use a linear combination between the upsampled values $z_{s-1}^{up}$ from the previous scale and the (shifted) outputs $z_s$ of the current scale, along with learned position embeddings *pos* at both locations. The input tokens to the transformer are then

$$inputs(s,i) = W_{proj} \cdot \text{Concat}(z_{s-1}^{up}[b_s^i], z_s[b_s^{i-1}], pos[b_s^i], pos[b_s^{i-1}]) \tag{1}$$

where $W_{proj}$ is a learned linear projection, and $[\cdot]$ indicates indexing at the specified block locations. Since the first block $b_s^1$ has no previous block, we use learned constant vectors for $z_s[b_s^0]$ and $pos[b_s^0]$.

At inference, each block $b_s^i$ contains the set of locations for each sampling step. Tokens are processed in parallel within each block, while the blocks themselves are serialized and sampled sequentially. Output tokens for each block are sampled independently using a multinomial.

During training, the entire sequence is processed in parallel using the ground truth codes $z_s$ for both inputs and targets autoregressively. We use a blockwise causal mask where tokens within each block $b_s^i$ can attend to each other, as well as to tokens from all previous blocks, both for the same scale and in lower-resolution scales. Each token position outputs softmax logits for a quantized codebook; we use flat mean cross-entropy loss over all scales and positions.

Importantly, the locations in each block are spaced as evenly as possible. The number of locations that can be sampled in parallel depends on the independence between locations, when conditioned both on the previous scale and samples in the current scale. Thus, there is a trade-off between the number of scales and the degree of parallelism within each scale, a relation that we explore later in Sec 4.3.

### 3.2 Progressive Scan Order

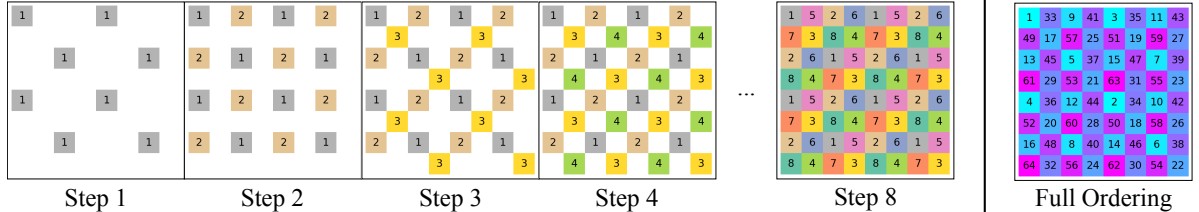

Figure 3: (a) Progressive checkerboard on an $8 \times 8$ grid using $P = 8$ steps. (b) Full ordering.

---

**Algorithm 1** Progressive Checkerboard Scan Order

```
def ProgressiveCheckerboard(size, x=0, y=0):
    if size == 1: return [(x, y)]
    d = size // 2
    # create balanced lists for TL, BR, TR, BL
    sublists = [
        ProgressiveCheckerboard(d, xi, yi)
        for (xi, yi) in ((x,y), (x+d,y+d), (x+d,y), (x,y+d))
    ]
    # combine round-robin from the quadrants
    return concat(zip(*sublists))
```

---

To create the progressive checkerboard ordering, we use a divide-and-conquer approach, where the 2D grid of size $N \times N$ is recursively subdivided into quadrants, and locations selected to maintain a spatially balanced assignment at all quadtree levels. At each recursion step, balanced index lists are generated for each sub-quadrant recursively, then merged using round-robin selection with a diagonal skip-step pattern (i.e., TL, BR, TR, BL). This results in a spatially balanced progressive order, with a unique index in $1 \dots N^2$ assigned to each of the $N \times N$ spatial positions. (If the grid size $N$ is not a power of 2, we generate the order on the next power of 2 and restrict to the grid size). See Algorithm 1.

This scan order can be used flexibly with various partition sizes. To create the blocks $b_s^i$ for each scale $s$, we simply divide the progressive scan order into $P$ contiguous segments of equal size. Figure 3 shows an example progressive checkerboard ordering on an $8 \times 8$ grid.

### 3.3 Token Embeddings

For token embeddings, we use quantized latent codes from a VAE-based autoencoder (Larsen et al., 2016; Esser et al., 2021; Ramesh et al., 2021; Rombach et al., 2022). Each scale is encoded separately from the

RGB image, so that latents are directly interpretable and decodable. For transformer inputs, we pass the VAE latent codes through a small MLP to unfold them from their low-dimensional clustering space back to higher dimension. In addition, we use a class token as the first input, which along with AdaLN (Peebles & Xie, 2023) is used to condition on class labels with classifier-free guidance (Ho & Salimans, 2022).

### 3.4 Position Encodings and RoPE Mixing

To make use of our blockwise checkerboard order's ability to model local dependencies, the transformer must be able to attend to adjacent locations for each token. We accomplish this using both input position embeddings and learned rotary encodings (RoPE) (Su et al., 2021; Heo et al., 2024). Embeddings encode the position in the image-scale space, not the sampling order. We initialize RoPE using factored space and scale representations (7/8 spatial, 1/8 scale). The embeddings are then learned in the joint space, allowing the model to find a high-frequency adjacent-position basis and other attention patterns optimized for our sampling order.

Each token in the transformer processes inputs from two locations: the current sampling positions $b_s^i$, and the conditioning latents at the previous block's positions $b_s^{i-1}$ (Sec. 3.1). Eq. (1) supplies both in the input layer mixture. However, without modification, the attention layers only use RoPEs for the current positions $b_s^i$. To enable attention on both sets of positions, we looked at mixing RoPEs for attention keys using learned mixing coefficients[1] $\alpha_{lh}$ for each transformer layer $l$ and attention head $h$:

$$rope_{lh}(b_s^i) = \alpha_{lh} \cdot rope(b_s^i) + (1 - \alpha_{lh}) \cdot rope(b_s^{i-1}),$$

where $rope(\cdot)$ produces RoPE embeddings for specified locations. Mixing is only applied to keys; queries always use the positions $b_s^i$, corresponding to the current sampling locations whose output is being computed. Although we did not see any performance gains with this strategy, we examine its behavior in Sec. 4.4, finding that only the first two layers use the previous block's RoPE shifts. This indicates that conditional information is extracted early on, so input mixing is sufficient.

## 4 Experiments

### 4.1 Model and Training Details

We train our models on ImageNet (Russakovsky et al., 2015) at 256x256 resolution, using two model sizes: the small (S) model has 12 transformer layers with hidden dimension 512 and 16 attention heads, while the large (L) model has 20 layers with 1024 hiddens and 16 heads. In most ablation experiments we use the S model, trained for 100 epochs with batch size 128 and learning rate $1 \times 10^{-4}$, dropped to $1 \times 10^{-5}$ in two steps over 10 epochs. The L model is trained for 200 epochs with batch size 64 and learning rate $5 \times 10^{-5}$, dropped to $1 \times 10^{-5}$ for 5 epochs, followed by a second effective drop by increasing batch size to 320 via gradient accumulation for the last 5 epochs. In all cases, we use AdamW (Loshchilov & Hutter, 2019) optimizer with weight decay 0.01 and ten-crop transforms, running on a single NVIDIA GH200 GPU.

We use the VAE autoencoder from LlamaGen (Sun et al., 2024). To better represent smaller image sizes in our multiscale model, we fine-tune by freezing all layers other than the quantizer codebook itself, and retrain the just the codebook layer with a size of 4096 on ImageNet for one epoch, using random image sizes between 16 and 256 and L2 loss on the latent codes. To measure sample quality, we compute Frechet Inception Distance (FID) (Heusel et al., 2017) and Inception Score (IS) (Salimans et al., 2016), using 50k samples and standard reference set using the TensorFlow implementation from Sun et al. (2024); Dhariwal & Nichol (2021).

For classifier-free guidance (CFG), we found applying CFG too early in the sampling progression limits diversity, likely due to the first samples' influence on global structure. Because of this, we sample the first 5 steps (corresponding to the 1x1 and 2x2 scales when scaling by 2x) with CFG=0, and apply CFG for all subsequent steps. Following common practice (Wang et al., 2025; He et al., 2025; Zhang et al., 2025), we perform a sweep of CFG values at 0.1 increments using the reference set.

---

[1]We learn the "logits" $\alpha'$ of the mixing coefficients and set $\alpha_{lh} = \text{sigmoid}(\alpha'_{lh})$.

| Model | Type/Tok | Params | FID ↓ | IS ↑ | Pre.↑ | Rec.↑ | #Steps | Time (s) |
|---|---|---|---|---|---|---|---|---|
| DiT-XL/2 (Peebles & Xie, 2023) | Diffu-KL | 675M | 2.24 | 278.2 | 0.83 | 0.57 | 1 × 250 | 11.9 |
| MAR-L (Li et al., 2024) | MAR-KL | 479M | 1.78 | 296.0 | 0.81 | 0.60 | 64 × 100 | 26.4 |
| GtR (Yan et al., 2025) | MAR-KL | 479M | 1.81 | 297.4 | — | — | 32 × 30 | — |
| xAR (Ren et al., 2025b) | Flow-KL | 608M | 1.28 | 292.5 | 0.82 | 0.62 | 4 × 50 | 7.7 |
| LlamaGen-L (Sun et al., 2024) | AR-VQ | 343M | 3.07 | 256.1 | 0.83 | 0.52 | 576 | 12.58 |
| VAR-d16 (Tian et al., 2024) | VAR-VQ | 310M | 3.30 | 274.4 | 0.84 | 0.51 | **10** | **0.12** |
| PAR-L-4x (Wang et al., 2025) | AR-VQ | 343M | 3.76 | 218.9 | 0.84 | 0.50 | 147 | 3.38 |
| RandAR-L (Pang et al., 2025) | AR-VQ | 343M | 2.55 | 288.8 | 0.81 | 0.58 | 88 | 1.97 |
| NAR-L (He et al., 2025) | AR-VQ | 372M | 3.06 | 263.9 | 0.81 | 0.53 | 31 | 1.01 |
| ARPG-L (Li et al., 2025) | AR-VQ | 320M | **2.30** | 297.7 | 0.82 | 0.56 | 32 | 0.58 |
| LPD-L (Zhang et al., 2025) | AR-VQ | 337M | 2.40 | 284.5 | 0.81 | 0.57 | 20 | 0.28 |
| Checkerboard-L 2x cfg=1.4 (Ours) | AR-VQ | 343M | 2.72 | **302.5** | 0.81 | 0.56 | 17 | 0.52 |
| Checkerboard-L 2x cfg=1.5 (Ours) | AR-VQ | 343M | 2.83 | **318.2** | 0.82 | 0.57 | 17 | 0.52 |
| Checkerboard-L 4x cfg=1.7 (Ours) | AR-VQ | 343M | 2.79 | **311.5** | 0.80 | 0.57 | 17 | 0.52 |

Table 1: Image generation models on ImageNet 256x256. We mainly compare to other AR-VQ methods (bottom rows) that are directly comparable, but also include non-VQ methods for additional context (top, × indicates mask/ar and diffusion/flow steps). Inference time measured for single image generation on A100.

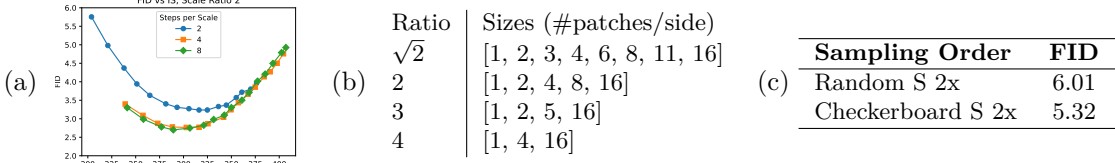

(a)     (b)

| Ratio | Sizes (#patches/side) |
|---|---|
| $\sqrt{2}$ | [1, 2, 3, 4, 6, 8, 11, 16] |
| 2 | [1, 2, 4, 8, 16] |
| 3 | [1, 2, 5, 16] |
| 4 | [1, 4, 16] |

(c)

| Sampling Order | FID |
|---|---|
| Random S 2x | 6.01 |
| Checkerboard S 2x | 5.32 |

Figure 4: (a) FID vs IS computed at 0.1 CFG increments. L model size at 2, 4 and 8 steps per scale. (b) Exact scale sizes used in our experiments, for 256x256 image and 16x16 VAE patch size. (c) Checkerboard vs random order, S model at 2x scale ratio. Checkerboard performs better due to its spatially balanced order.

## 4.2 Benchmark Comparison

Table 1 compares our method against recent autoregression-based image generation models of similar model size on ImageNet 256x256. We show results for our Checkerboard-L model with a 2x scaling ratio and 4 sampling steps per scale (17 total), as well as a 4x ratio with 8 steps per scale (also 17 total)[2]. Directly relevant comparisons are with other autoregression-based methods with VQ sampling, shown with "AR-VQ" in the second column. We also show methods that use non-quantized autoencoders for wider context, though these are not directly comparable. Qualitative samples are in Fig. 8, and selected failures in Appendix C.

Compared to PAR (Wang et al., 2025) and RandAR (Pang et al., 2025), to which our method is most closely related, we achieve *similar or better FID and IS with fewer sampling steps and faster inference time*. While PAR uses 147 steps and RandAR 88 steps, we only need 17 steps to achieve a FID of 2.72 (compared to 3.76 and 2.55, respectively). Our inference time of 0.52s per image is also faster than both methods (3.38s for PAR and 1.97s for RandAR) on A100.

In addition, our method is competitive with the more recent AR-based models ARPG (Li et al., 2025) and LPD (Zhang et al., 2025), notably also using fewer steps, though FID is somewhat lower. Although the reasons for lower FID are unclear, two possibilities are: (*i*) Training uses batch size 64 for 200 epochs (increased to 320 for the last 5) while LPD and ARPG use 2048 and 1024, respectively, for 400 epochs. Thus, our schedule has fewer total examples and noisier gradients. And (*ii*) ARPG and LPD separate tokens for sampled values and queries, while we combine them in an input mixture. This results in half as many tokens during training, but at the cost of using some KV cache capacity for query information that may not be useful for later steps. Combining strengths of all three approaches is a promising direction for future work.

---

[2]For the L models, the 2x model is trained from scratch for 200 epochs, and the 4x model is initialized with the 2x model weights and trained for 30 epochs.

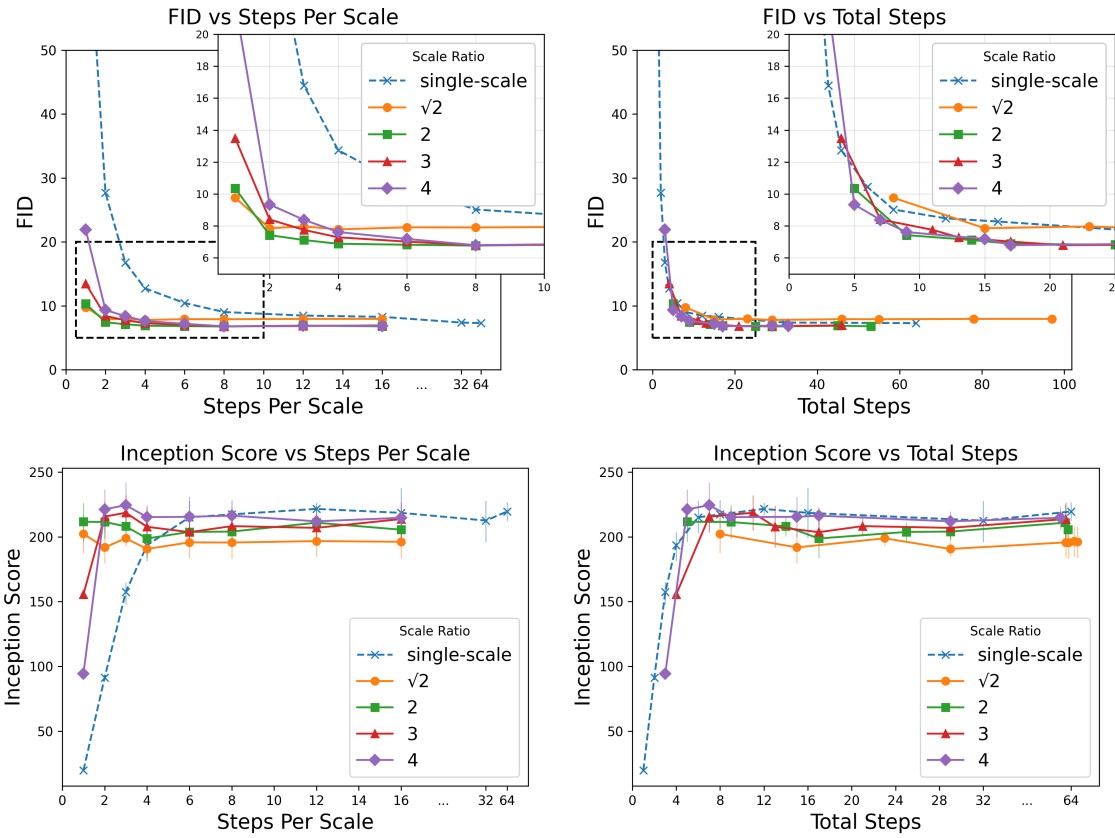

Figure 5: FID (top) and IS (bottom), by scale ratio and number of inference steps for the S model size. Left: by numbers of steps per scale; Right: by total number of steps. Multiscale models outperform the single-scale baseline. Note that scale ratios 2, 3 and 4 all achieve similar performance counted by *total* steps, even though the steps per scale needed to reach each total is different for each scale factor.

We also verify the effectiveness of our checkerboard ordering in Figure 4(c): using the S model at 2x scale ratio, we train a comparison model using random within-scale order. The checkerboard performs substantially better, due to its spatially balanced sampling pattern. In addition, we show the full CFG sweep for our L model in Fig. 4(a). While only 2 steps per scale is ineffective, both 4 and 8 perform similarly. In the next section, we explore the impact of number of steps in greater detail, and how these relate to the scaling ratio.

## 4.3 Relationship Between Scale Ratio and Sampling Steps

To evaluate and compare different scale factors and sampling steps, we train our small (S) sized model using four scale-up factors: $\sqrt{2}, 2, 3$ and 4, as well as a single-scale baseline (exact sizes in Fig 4(b)). We then evaluate each one using different numbers of sampling steps, varying the number of steps per scale, which corresponds to partitioning using the checkerboard pattern. Note that in our implementation, each scale ratio requires retraining, but the number of sampling steps can be varied at inference.

We evaluate each condition using FID and IS with 50,000 samples, sweeping classifier-free guidance in 0.1 increments to obtain the best FID value for each setting. CFG is applied at all scales. To account for small variations in evaluation and sampling, for each point we record values with FID within 2% (relative) of its minimum, and plot the mean, min and max IS of these with error bars.

Figure 5 shows the results. On the left, we plot FID and IS by number of steps per scale, while on the right we plot them by total number of steps. While IS scores are similar between multiscale models and the single-scale baseline, all multiscale models outperform the single-scale baseline in terms of FID, with the best performance achieved at scale ratios 2, 3 and 4. As expected, larger scale-up factors require more steps per scale to achieve good performance (left-side figures).

However, when viewed by the *total* number of steps, scale ratios 2, 3 and 4 all achieve similar performance, *even though each scale ratio uses a different number of steps per scale* to reach the total. The lines for these three ratios all overlap tightly (Fig. 5, top-right). In addition, all outperform the single-scale baseline as well as slower $\sqrt{2}$ scaling. This indicates that while multiscale conditioning is important, the exact scale factor is not very sensitive: there are multiple ways to allocate a given number of steps among scales and obtain similar performance. Indeed, our checkerboard order also includes a degree of coarse-to-fine conditioning, simply due to its spatially balanced sampling, which may contribute to this effect (see Appendix E). Thus, for our balanced multiscale generation, *the total number of steps in the conditional chain is the dominant performance factor*.

Moreover, best performance is achieved at around 17 total steps for all of the multiscale models, while steps beyond this add little if any benefit. This makes sense, since partitioning our balanced checkerboard order into more steps spaces out the locations within each step, and coarse-to-fine conditioning is also included in the multiscale pyramid. Within-scale steps are needed largely to condition within the local upsampling windows. However, while between- and within-scale conditioning can model overlapping dependencies, we still find benefit to including both.

Additionally, we verify our findings at 512 resolution in Appendix A, and for the L model at 256 in Appendix B.

## 4.4 RoPE Mixing

As described in Sec. 3.3, we experimented with mixing RoPE embeddings for attention keys using learned mixing coefficients $\alpha_{lh}$ for each layer $l$ and head $h$. Table 2(a) compares mixing for all layers, no layers, and only the first two layers. Training time increases when enabled for all layers, but is negligible when applied just to the first two. While we found no significant differences in FID or IS, the behavior of the mixing coefficients shown in 2(b) is illustrative of how the model uses sampling information in the inputs. Only the first two layers attend to sampled positions, indicated by negative weights. This suggests the model may extract information from the sampled locations early, possibly embedding it into representations at relevant output positions, and relying on attention based on output positions for most of the transformer. While this hypothesis is speculative, it is clear only the first layers use sampled positions, so including these in the input as in Eq. (1) is sufficient.

(a)

| **RoPE Mixing** (S model size) | | | |
|---|---|---|---|
| | s/Batch | FID ↓ | IS ↑ |
| No Mixing | 0.181 | 5.32 | 234.7 |
| All Layers | 0.195 | 5.33 | 230.2 |
| First 2 Layers | 0.184 | 5.46 | 226.5 |

(b)

Table 2: RoPE mixing strategies (see Sec. 3.3). (a) Results comparison; times use batch size 128 on single GH200. (b) Mixing weights when enabled for all layers. Negative weights (blue) correspond to sampled positions, while positive (red) correspond to output locations. Only the first two layers have negative weights: Information for previously-sampled values is extracted early.

## 4.5 Entropy Analysis

To further illustrate the behavior of multiscale checkerboard sampling, we plot measurements of entropy for the multinomial distribution at each token during sampling. Fig. 6 shows aggregate measurements taken over 10,000 random samples of our L model. We show mean entropy as well as regions corresponding to 25th and 75th percentiles, over each sampling step. Appendix D confirms statistical significance of the following descriptions. As expected, entropy decreases as steps progress and samples resolve modes and variance by conditioning. Between scales, though, there is a jump in entropy, as additional higher-resolution details are introduced, followed by continued decrease as sampling within the scale develops. Interestingly, on average, the largest entropy drop within each scale occurs half-way through the ordering, when every-other location has been filled. This is the transition point where each unsampled location has 4 adjacent sampled neighbors.

For comparison, we also plot entropy for PAR and RandAR. PAR samples 4 quadrants in parallel with raster order and has a sawtooth pattern corresponding to the raster rows. The last row shows a drop, since the bottom row of top half is adjacent to already-sampled rows of the bottom half. RandAR samples in a random order; correspondingly, its entropy smoothly decreases on average, corresponding to its schedule.

Fig. 7 shows entropy for individual samples from our method, both over steps, and over locations. Here, we see a trend consistent with the aggregate measures, but also substantial variation, since the token distribution depends heavily on the image content being generated. Additionally, the entropy maps clearly display checkerboard patterns, a result of our sampling order. This underscores our method's ability to model local conditional dependencies, particularly between adjacent locations.

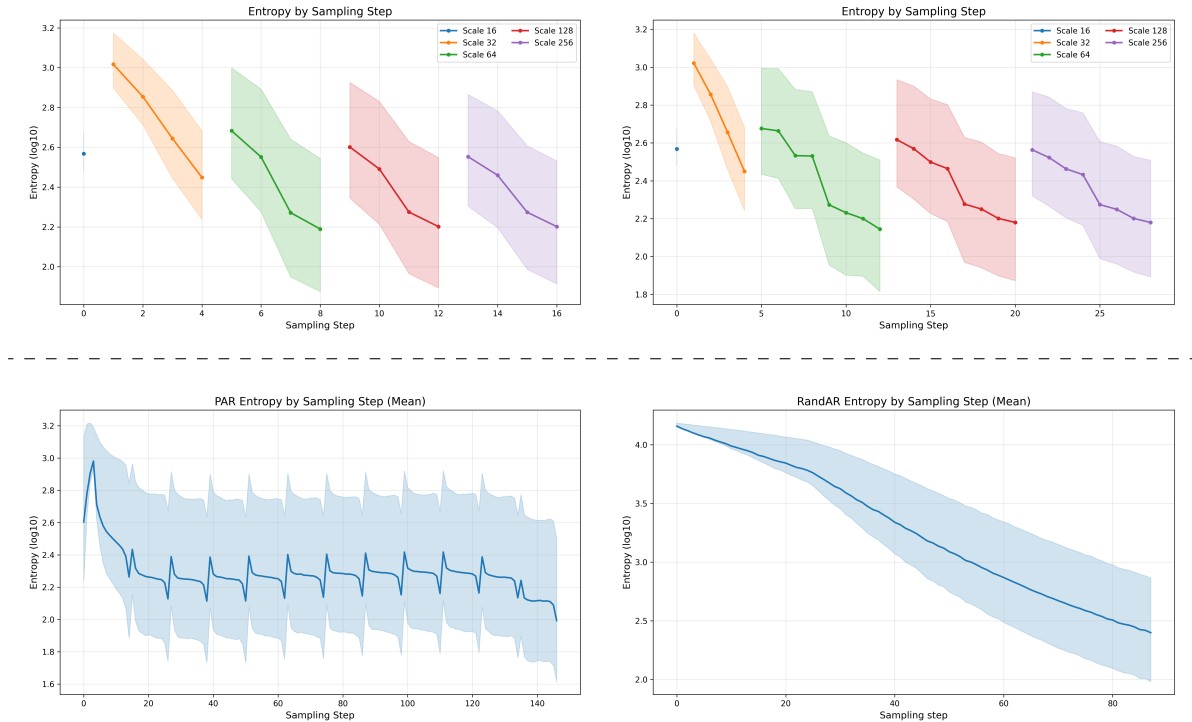

Figure 6: Top: Aggregate entropy measurements over 10K samples. Entropy decreases within each scale, but jumps between scales as new details are introduced. Left: 4 steps/scale; Right: 8 steps/scale. Bottom: Entropy measured for PAR and RandAR, for comparison.

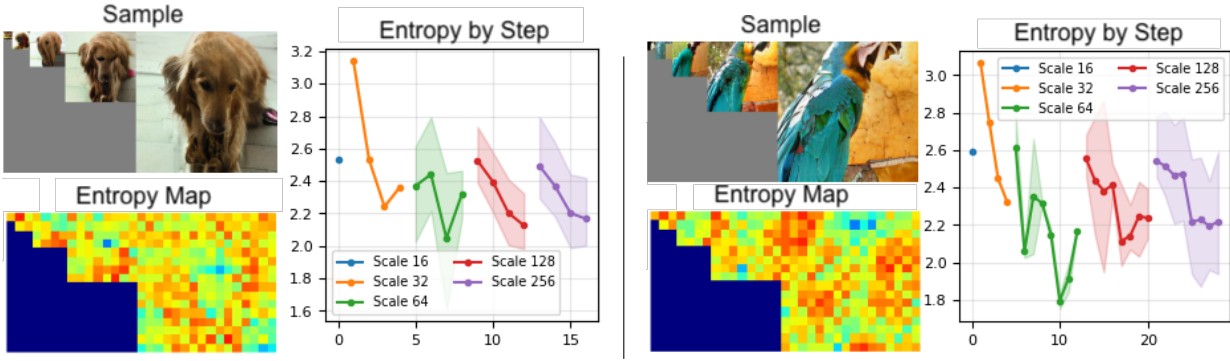

Figure 7: Entropy measurements for individual samples. Left: 4 steps/scale; Right: 8 steps/scale.

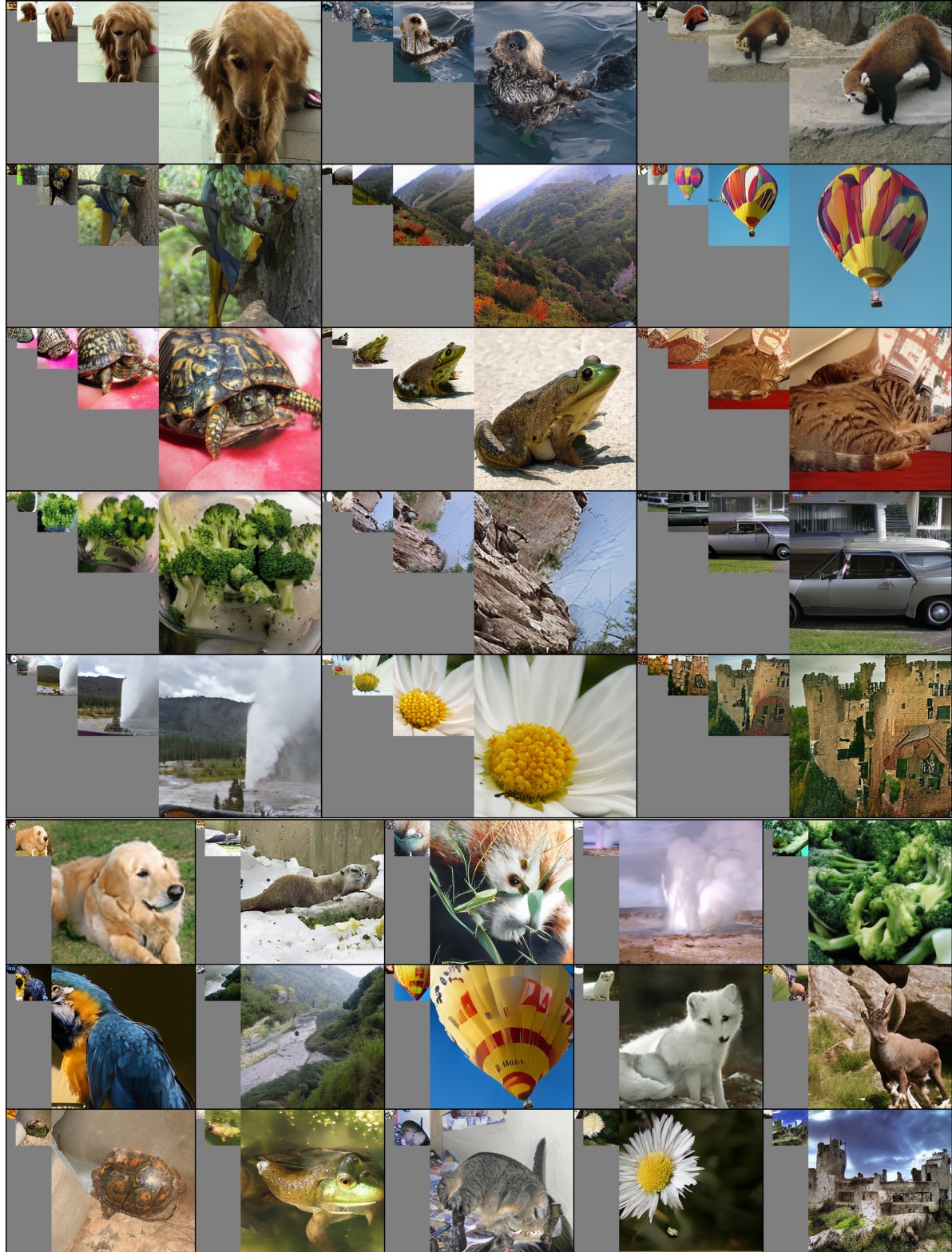

Figure 8: Samples from our Checkerboard-L model, scale factor 2x (top) and 4x (bottom).

## 5 Conclusion

We have described a multiscale image generation method based on a spatially-balanced progressive checkerboard sampling order. By modeling both between-scale and within-scale conditional dependencies, our method is able to generate images with fewer sampling steps and competitive inference time compared to recent methods. In this setting, our experiments on ImageNet show that several multiscale pyramid ratios lead to similar performance for a given number of total sampling steps, enabling our method to use large scaling ratios of up to 4x. Future work may include extensions to video or text modalities, or further studying the relationship between scales and steps in the context of mutual information and dependency modeling.

## 6 Broader Impacts

Image generation has a wide range of applications, including creative content generation and data augmentation for training other models. It also has potential risks, including misuse by generating deepfakes or other deceptive content, surfacing biases in training data and perpetuating inaccurate representations, including when used for generating new training data. Additionally, improved efficiency, especially in an autoregressive model close to those used for text generation, may enable widespread multimodal reasoning models, which could have significant impacts, such as improved reasoning through visual sketches, and tighter automatic visual refinement loops; these may enable new unforeseen applications in turn, with potential effects both positive and negative.

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

## Appendix A: Experiments at 512x512 Resolution

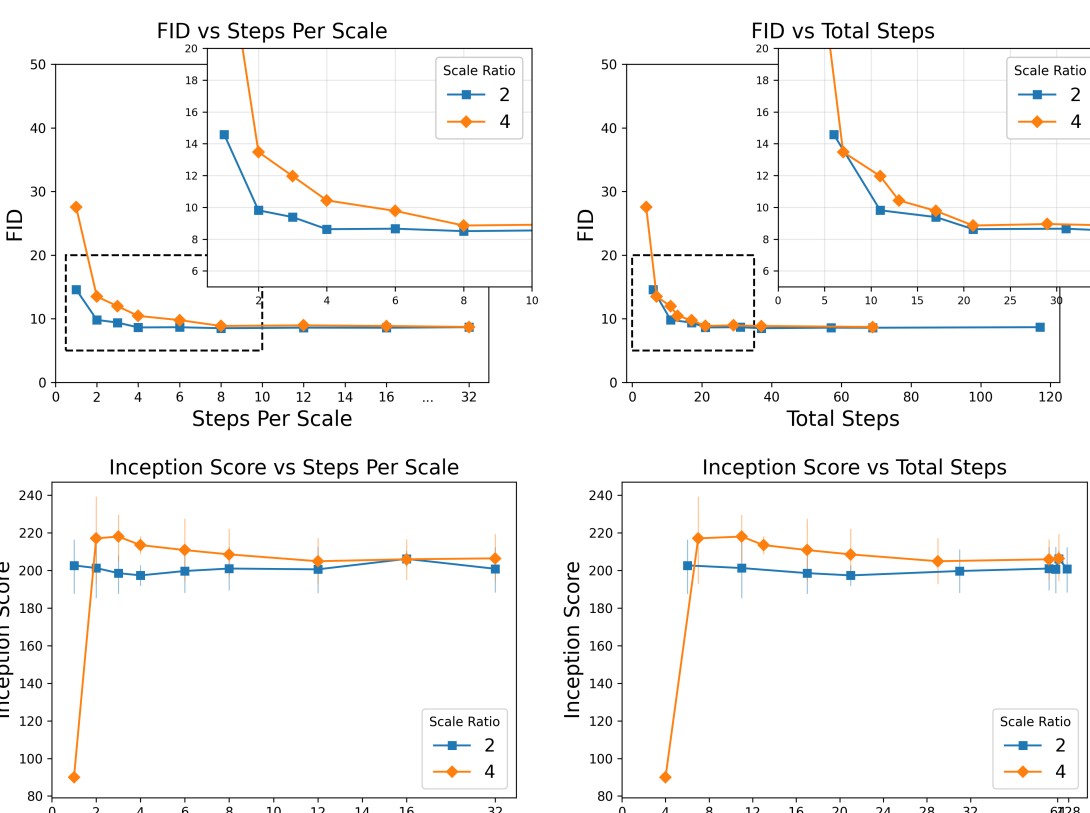

Figure 9: FID and IS by scale ratio and number of steps for the S model, at $512 \times 512$ resolution using incremental transfer learning from the corresponding models trained at 256 resolution (see text for details). Left: by numbers of steps per scale; Right: total number of steps.

To investigate whether our findings regarding the relationship between scale ratio and sampling steps generalize to higher resolutions, we perform a limited set of experiments at $512 \times 512$ resolution.

Using the small (S) model size, we train two models at $512 \times 512$ resolution, using 2x and 4x scale ratios. Each are initialized using weights from the corresponding model trained at 256 resolution, and then trained using a short transfer schedule for 10 epochs (1 warmup $\rightarrow$ 5 main lr $\rightarrow$ 4 decreased lr with 2 stepdowns).

Note the 2x model simply appends the 32 scale to its pre-existing progression: $[1, 2, 4, 8, 16, 32]$. However, the 4x model has to change all scales in its list, from $[1, 4, 16]$ to $[1, 2, 8, 32]$, so is not as natural a transfer setting.

The results are shown in Fig. 9, largely supporting our findings from the 256 resolution experiments. Viewed by total steps, the models' FID curves are close to one another, both reaching best FID at 21 total steps. In this case, the 2x model is slightly better in FID than the 4x model. However, it's unclear if this is due to fundamental differences or experimental limitations, since the 4x model's scale progression is less natural in this transfer setting. Overall, the results suggest that while the exact scale ratio may start to have some more impact on performance at 512 resolution, the total number of steps is still a dominant factor, and that multiple scale ratios can achieve similar performance when using the same total number of steps.

## Appendix B: Total Steps Evaluations for L Model

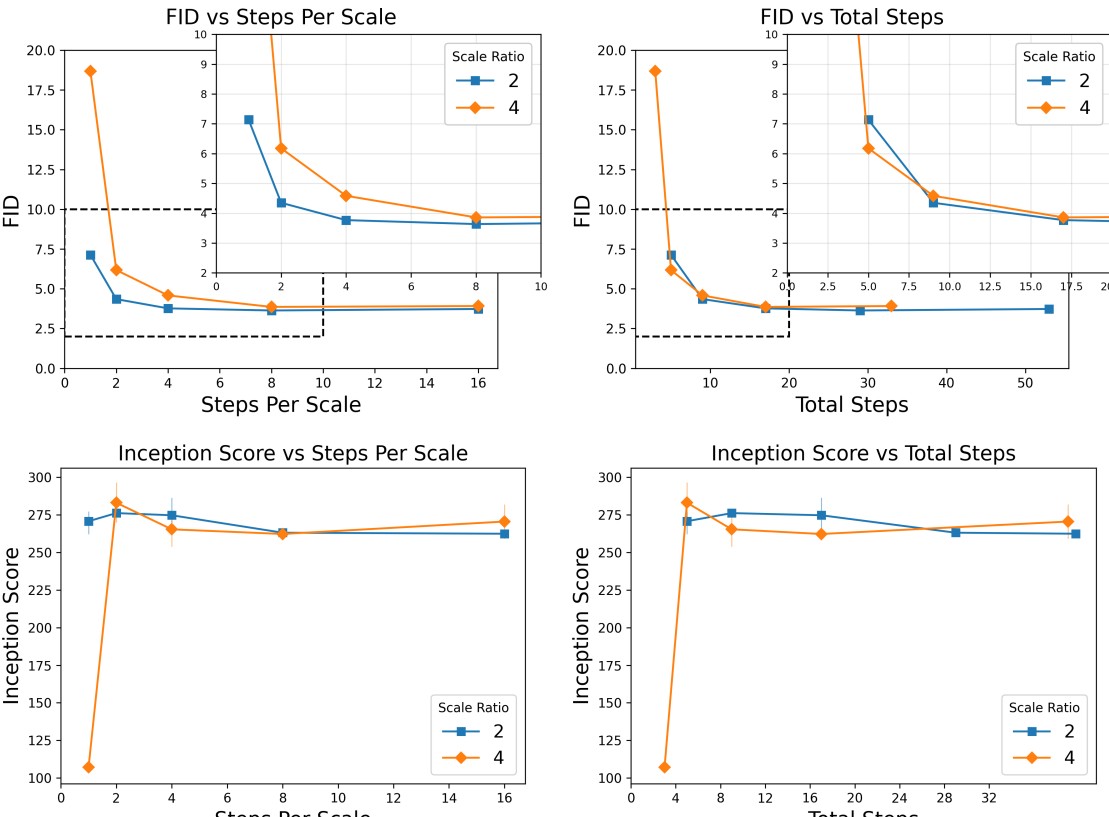

Figure 10:   FID and IS by scale ratio and number of steps for the L model (256 resolution).

Using the large (L) model, we plot FID and IS by scale ratio and number of steps in Fig. 10.

The results are consistent with those from the S model, showing that while 4x scaling requires more steps per scale than 2x, when viewed by total steps, performance is similar between the two. While the 2x ratio may achieve slightly better FID, the total number of steps is still dominant, with best FID achieved at 17 total steps for both ratios.

## Appendix C: Qualitative Failure Cases

Below we show some qualitative failure cases from our Checkerboard-L model at 256 resolution, using 4 steps per scale. Although none can be linked conclusively to artifacts from checkerboard sampling, cases of medium- and long-range object distortion, particularly of highly geometric objects like circles and straight lines, might be influenced by this if token samples are out of place from their long-range alignment.

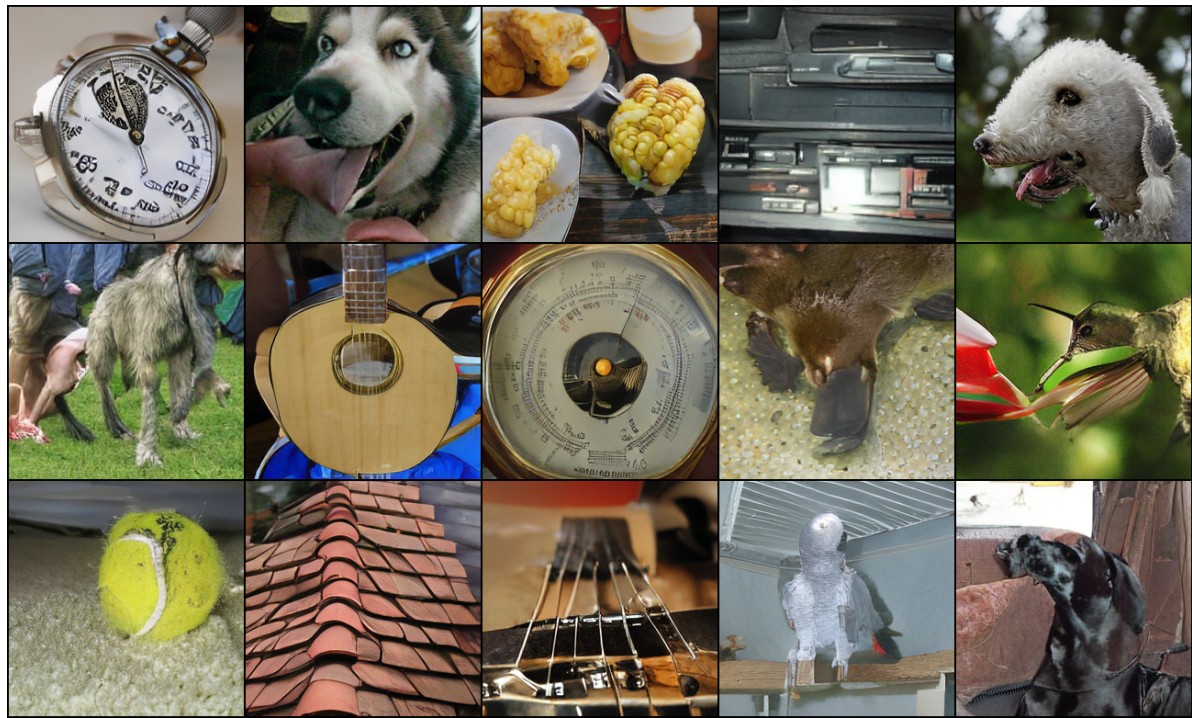

Figure 11: Qualitative failure cases from our Checkerboard-L model, scale factor 2x.

## Appendix D: Entropy Analysis Statistics

As described in Sec. 4.5, entropy decreases within each scale on average, with the average largest decrease half-way through the ordering, when every-other location has been filled. Here we confirm the statistical significance of these observations.

First, we check the entropy decrease for each sampling step. For $P = 4$ steps per scale, entropy is on average lower in every step $t$ compared to step $t − 1$ ($t \leq −28.2$; $p < 10^{-100}$). This is also the case for the 8x8 and 16x16 scales with $P = 8$ ($t \leq −11.7$; $p < 10^{-30}$). For the 4x4 scale with $P = 8$, only two locations are sampled per step, and the decrease is significant at all but two steps, visible as flat lines for this scale in Fig 6 (right side, green line). Full results are in Table 3 below.

Next, we check the entropy drop at the half-way point is the largest drop as described, comparing its size compared to the next-largest drop in each scale. Both $P = 4$ and $P = 8$ steps per scale have signficantly larger decreases at their half-way points ($t \leq −24.1$; $p < 10^{-100}$). Full results are in Table 4.

| Scale | | $P = 4$ | | | | | $P = 8$ | | | |
|---|---|---|---|---|---|---|---|---|---|---|
| | $t$ | Mean $\Delta H$ | SE | SD | $t$-stat | $t$ | Mean $\Delta H$ | SE | SD | $t$-stat |
| | 1 | $-0.315$ | 0.006 | 0.607 | $-51.9$ | 1 | $-0.017$ | 0.007 | 0.713 | $-2.4^{\ddagger}$ |
| | 2* | $-0.661$ | 0.006 | 0.597 | $-110.9$ | 2 | $-0.324$ | 0.008 | 0.766 | $-42.3$ |
| | 3 | $-0.182$ | 0.006 | 0.644 | $-28.2$ | 3 | $-0.009$ | 0.007 | 0.748 | $-1.2^{\ddagger}$ |
| $4 \times 4^{\dagger}$ | | | — | | | 4* | $-0.594$ | 0.008 | 0.788 | $-75.4$ |
| | | | — | | | 5 | $-0.111$ | 0.008 | 0.812 | $-13.7$ |
| | | | — | | | 6 | $-0.063$ | 0.008 | 0.838 | $-7.5$ |
| | | | — | | | 7 | $-0.130$ | 0.008 | 0.813 | $-16.0$ |
| | 1 | $-0.256$ | 0.003 | 0.308 | $-83.0$ | 1 | $-0.116$ | 0.003 | 0.342 | $-34.0$ |
| | 2* | $-0.512$ | 0.003 | 0.297 | $-172.2$ | 2 | $-0.170$ | 0.004 | 0.372 | $-45.7$ |
| | 3 | $-0.167$ | 0.003 | 0.318 | $-52.7$ | 3 | $-0.076$ | 0.004 | 0.351 | $-21.6$ |
| $8 \times 8$ | | | — | | | 4* | $-0.439$ | 0.004 | 0.386 | $-113.8$ |
| | | | — | | | 5 | $-0.053$ | 0.004 | 0.377 | $-14.0$ |
| | | | — | | | 6 | $-0.118$ | 0.004 | 0.405 | $-29.2$ |
| | | | — | | | 7 | $-0.044$ | 0.004 | 0.375 | $-11.7$ |
| | 1 | $-0.214$ | 0.001 | 0.148 | $-144.4$ | 1 | $-0.094$ | 0.002 | 0.157 | $-60.0$ |
| | 2* | $-0.435$ | 0.001 | 0.147 | $-296.3$ | 2 | $-0.141$ | 0.002 | 0.179 | $-78.7$ |
| | 3 | $-0.163$ | 0.002 | 0.151 | $-107.9$ | 3 | $-0.071$ | 0.002 | 0.161 | $-44.1$ |
| $16 \times 16$ | | | — | | | 4* | $-0.370$ | 0.002 | 0.184 | $-200.9$ |
| | | | — | | | 5 | $-0.054$ | 0.002 | 0.172 | $-31.5$ |
| | | | — | | | 6 | $-0.112$ | 0.002 | 0.193 | $-57.9$ |
| | | | — | | | 7 | $-0.044$ | 0.002 | 0.173 | $-25.3$ |

Table 3: Mean adjacent-step entropy change $\Delta H$ (nats), for $P = 4$ and $P = 8$ steps/scale ($N = 10,000$ samples, CFG=1.5). *marks the midpoint step. All $t$-statistics correspond to $p < 10^{-10}$ (one-tailed $t$-test), except where marked $^{\ddagger}$ (not significant).

| Scale | | $P = 4$ | | | | | $P = 8$ | | | |
|---|---|---|---|---|---|---|---|---|---|---|
| | vs. $t$ | Mean $\delta$ | SE | SD | $t$-stat | vs. $t$ | Mean $\delta$ | SE | SD | $t$-stat |
| | 1 | $-0.346$ | 0.010 | 1.030 | $-33.6$ | 1 | $-0.577$ | 0.011 | 1.080 | $-53.5$ |
| | 2* | 0 | — | — | — | 2 | $-0.270$ | 0.011 | 1.119 | $-24.2$ |
| | 3 | $-0.480$ | 0.011 | 1.089 | $-44.1$ | 3 | $-0.585$ | 0.013 | 1.319 | $-44.4$ |
| $4 \times 4$ | | | — | | | 4* | 0 | — | — | — |
| | | | — | | | 5 | $-0.483$ | 0.014 | 1.396 | $-34.6$ |
| | | | — | | | 6 | $-0.531$ | 0.012 | 1.175 | $-45.2$ |
| | | | — | | | 7 | $-0.464$ | 0.012 | 1.147 | $-40.4$ |
| | 1 | $-0.256$ | 0.005 | 0.522 | $-49.1$ | 1 | $-0.323$ | 0.005 | 0.523 | $-61.7$ |
| | 2* | 0 | — | — | — | 2 | $-0.269$ | 0.006 | 0.559 | $-48.1$ |
| | 3 | $-0.344$ | 0.005 | 0.543 | $-63.3$ | 3 | $-0.363$ | 0.006 | 0.631 | $-57.5$ |
| $8 \times 8$ | | | — | | | 4* | 0 | — | — | — |
| | | | — | | | 5 | $-0.386$ | 0.007 | 0.658 | $-58.7$ |
| | | | — | | | 6 | $-0.321$ | 0.006 | 0.582 | $-55.1$ |
| | | | — | | | 7 | $-0.395$ | 0.006 | 0.546 | $-72.3$ |
| | 1 | $-0.221$ | 0.003 | 0.248 | $-89.4$ | 1 | $-0.276$ | 0.002 | 0.239 | $-115.4$ |
| | 2* | 0 | — | — | — | 2 | $-0.230$ | 0.003 | 0.272 | $-84.3$ |
| | 3 | $-0.272$ | 0.003 | 0.257 | $-106.2$ | 3 | $-0.299$ | 0.003 | 0.290 | $-103.1$ |
| $16 \times 16$ | | | — | | | 4* | 0 | — | — | — |
| | | | — | | | 5 | $-0.316$ | 0.003 | 0.303 | $-104.3$ |
| | | | — | | | 6 | $-0.259$ | 0.003 | 0.283 | $-91.5$ |
| | | | — | | | 7 | $-0.327$ | 0.003 | 0.252 | $-129.4$ |

Table 4: Difference between half-way step and every other step, $\delta = \Delta H_{\mathrm{mid}} − \Delta H_t$ (nats). $\delta < 0$ at every comparison confirms the middle step has the largest entropy change (one-tailed $t$-test), all $t$-statistics are significant with $p < 10^{-100}$.

## Appendix E: Further Discussion on Relationship Between Scale Ratio and Sampling Steps

In this section, we provide additional discussion on possible reasons why the scale-steps relationship shown in Sec. 4.3 arises.

Consider the following $4 \times 4$ area, with four of the 16 locations labeled for reference:

$$
\begin{array}{cccc}
A & \cdot & B & \cdot \\
\cdot & \cdot & \cdot & \cdot \\
C & \cdot & D & \cdot \\
\cdot & \cdot & \cdot & \cdot
\end{array}
$$

The question here is, how many locations can we sample independently?

First, consider a 2x scaling ratio. When scaling up, each pixel at scale $i-1$ "turns into" a $2 \times 2$ window of pixels in scale $i$. In the example above, this means that we've already sampled four values in scale $i-1$, corresponding to the $2 \times 2$ locations anchored at A, B, C and D. So for the most part, A, B, C and D are conditionally independent of each other, given samples at the corresponding locations in the previous scale. In practice, this is not entirely true — there could be fine-grained textures that need to be consistent between A and B in the new scale, for example — but it's mostly the case. Thus we can sample all four at the same time.

In general, this suggests that when upsampling with a ratio S, so that each pixel at scale $i-1$ "turns into" a window of $S \times S$ pixels at scale $i$, we can sample each window in parallel, but the $S \times S$ locations within each window may still need to be sampled sequentially.

For a $S = 2$ ratio, corresponding to $S \times S = 4$, this would imply 4 steps/scale, which is what we see in practice.

For a $S = 4$ ratio, corresponding to $S \times S = 16$, this would imply 16 steps/scale. However, as shown by our measurements in Sec. 4.3, we can use only 8 and get similar results. That is, A and D in the above grid can be sampled at the same time, even though they are within the same $S \times S$ window. Thus, it's possible to view the 8 steps/scale as an approximation of the 16 we know are sufficient according to the conditional independence induced by upsampling.

A key enabler here is very likely information redundancy: that is, information provided by some samples may overlap with the information provided by other samples. In particular, the information provided from samples of A,B,C,D above using $S = 4$x upsampling overlaps information that would have been supplied in the previous scale using $S = 2$x upsampling. This is because of the balanced sampling order, which spaces out samples uniformly in the image area. Once A, B, C and D are sampled, we are already conditioning on some information from each $2 \times 2$ window, even when scaling up by 4x: part of the information that would have been provided in each $2 \times 2$ window is also part of the A, B, C and D sampled values. The redundancy in the information enables more aggressive parallel sampling than might be predicted by the sequential-within-each-window assumptions — how much more parallelism is enabled by this mechanism depends on the data.

This information redundancy provides a basis for starting to understand why different scale ratios achieve similar quality at the same total step count. The balanced sampling order partially compensates for larger upsampling ratios, making the system insensitive to the exact ratio used, so that within a range of scale factors, performance is largely determined by the total number of steps.

