# OpenReview forum: "Progressive Checkerboards for Autoregressive Multiscale Image Generation"
_TMLR — Accepted by TMLR_

### Review · Reviewer_GqeC · 2026-04-01

**Summary Of Contributions:**

This paper proposes a spatially-balanced progressive checkerboard ordering for multiscale autoregressive image generation. The method combines between-scale and within-scale conditioning using a quadtree-based scan order, achieving competitive FID/IS on ImageNet 256×256 with only 17 sampling steps. The  paper finds that total serial steps, not the specific scale factor, governs performance.

The paper is well-written, cleanly structured. However, the contribution sits somewhat between engineering elegance and conceptual depth: the ordering itself is straightforward, and the experimental scope is limited to class-conditional ImageNet at a single resolution with a single tokenizer.

**Additional Comments:**

NA

**Audience:**

Yes

**Audience Explanation:**

1) The total-steps finding proposed in the paper is a surprising empirical result about autoregressive modeling. Researchers working on autoregressive models in any modality, not just images, would find this relevant, since it speaks to fundamental questions about how conditional dependencies are structured in sequential generation.

2) The ordering design works like a trade-off between parallelism and conditional modeling. It is central to efficient inference in any autoregressive system. The progressive checkerboard method is an analyzable example that could inspire analogous work in other domains.

**Claims And Evidence:**

Yes

**Claims Explanation:**

The progressive checkerboard ordering is simple, elegant, and formally well-motivated. Its quadtree-based construction ensures spatial balance at all levels, enabling flexible parallelism by simply varying the partition size P. The paper finds that scale ratios 2, 3, and 4 yield overlapping FID curves when plotted by total steps. In the experiments part, the authors show the efficiency of the proposed method, achieving FID 2.72 with only 17 steps. The inference time of 0.52s on A100 is also competitive.

**Requested Changes:**

1) All experiments are conducted on ImageNet 256×256 with a single tokenizer. No experiments on higher resolutions, text-to-image generation, or alternative tokenizers. The claim about scale-step interchangeability may not generalize. Higher resolutions and different tokenizers may exhibit different dependency structures. I suggest the authors to include at least one experiment at 512×512 or with a different tokenizer to demonstrate generality.

2) Training on a single GPU limits the model sizes explored. The L model has only 343M parameters. SOTA methods like Infinity and STAR explore much larger models. It is unclear whether the checkerboard ordering's advantages persist at scale.

3) The entropy analysis (Figs. 6–7) is presented as aggregate statistics (mean, 25th/75th percentiles) over steps, but lacks formal hypothesis testing or information-theoretic quantities like mutual information between adjacent tokens. The paper claims the entropy patterns "underscore our method's ability to model local conditional dependencies". A comparison of entropy profiles between the checkerboard ordering and a baseline (e.g., raster order) would be much more informative.

4) Missing comparison with Infinity at comparable scale. Infinity uses BSQ tokenization with bitwise self-correction and achieves strong results with ≈1.26× scale factors. While it is cited in related work, there is no experimental comparison at a comparable model size in Table 1.

5) The paper shows successful samples (Fig. 8) but does not analyze failure cases. At only 17 steps, there may be characteristic artifacts from insufficient conditioning. Understanding when and how the method fails is critical for practical adoption. I suggest the authors to show some cherry-picked failure cases and analyze their relationship to the checkerboard ordering.

---

> ### Author Response · Authors · 2026-04-27
> **discussion responses**
>
> Thank your for your detailed review and comments.  We have uploaded a new revision of the manuscript, with new analyses, described in our replies below.
>
> **Larger model sizes, single-gpu training**
>
> This work is the result of a self-funded independent project, and so we are not
> able to train larger sizes due to resource limitations.  Our choice to train
> single GPU was purely due to single GH200 being most cost-efficient.  Multi-GPU
> training also works, and in fact we used it while determining costs.
> We agree studying scaling properties on larger sizes would be a natural step,
> but is beyond our budget.
>
>
> **Experiment at 512×512 image size**
>
> Thank you for the suggestion.  We were able to run a limited experiment at 512
> image size within our budget, to verify the relation between steps and scale at
> this resolution.
>
> Appendix A in the new revision shows the results of this experiment.
>
> Using the small (S) model size, we train two models at $512\times512$ resolution, using 2x
> and 4x scale ratios.  Each are initialized using weights from the corresponding
> model trained at 256 resolution, and then trained using a short transfer
> schedule for 10 epochs (1 warmup $\rightarrow$ 5 main lr $\rightarrow$ 4
> decreased lr with 2 stepdowns).
>
> Note the 2x model simply appends the 32 scale to its pre-existing progression:
> $[1,2,4,8,16,32]$.  However, the 4x model has to change all scales in its list, from
> $[1,4,16]$ to $[1,2,8,32]$, so is not as natural a transfer setting.
>
> The results continue to support the relationship.  The models' FID curves are
> close when viewed by total steps, both reaching best FID at 21 total steps.
> The 2x model is slightly better in FID than the 4x model in this case.
> However, it's unclear if this is due to fundamental differences or experimental
> limitations, since the 4x model's scale progression is less natural in this
> transfer setting.
>
> Overall, the results suggest that while the exact scale ratio may start to have
> some more impact on performance at 512 resolution, the total number of steps is
> still a dominant factor, and that multiple scale ratios can achieve similar
> performance when using the same total number of steps.
>
>
> **Entropy analysis for other methods / orderings**
>
> This is a great suggestion.  We have added similar entropy analysis for PAR and
> RandAR in the new revision in Figure 6.
>
> PAR samples 4 quadrants in parallel, using raster order in each quadrant.  Its
> entropy profile shows an initial drop, followed by sawtooth patterns
> corresponding to the raster rows.  The very end shows another drop, as the
> final row in the top two quadrants are adjacent to the already-sampled
> rows of the bottom quadrants.
>
> RandAR samples in a random order, so its entropy profile is more smooth.  There
> are no sharp drops, with smooth decrease corresponding to the sampling
> schedule.
>
> Our multiscale checkerboard, in comparison, shows an entropy increase at the
> start of each scale, as new higher-resolution details need to be chosen.
> Within each scale, entropy decreases consistently, as the sampled tokens are
> spatially balanced.  The largest decrease happens in the middle of each scale, when
> every other location has been filled in: this is the transition point where
> each unsampled location has 4 adjacent sampled neighbors.
>
>
>
> **Infinity missing in Table 1**
>
> Unfortunately, there is no readily available comparison for Infinity that would
> be comparable in Table 1.  The paper only reports results on their
> text-conditioned dataset, and there are no trained weights released either for
> ImageNet or in the 300M param range.  The closest available are 125M or 2B,
> both trained on their text-conditioned dataset.  Thus we did not include it in
> Table 1.
>
>
> **Anecdotal failure cases**
>
> Thank you for the suggestion.  We have added some anecdotal failure cases in
> the new revision.  Although none can be linked conclusively to artifacts from
> our checkerboard sampling, we have found some cases of object distortion
> that could be influenced by this.  The new Appendix B shows examples.

---

### Review · Reviewer_h9Wv · 2026-04-09

**Summary Of Contributions:**

Though this is not my area of specialization, I understood this work to be a new technique for autoregressive multiscale image generation that generates images across scales (from coarse to fine), but with only carefully chosen subsets of locations filled in. The subsets themselves come from a novel progressive checkerboard ordering, making the locations closer to conditionally independent, so parallel sampling is safer. There are really multiple contributions:
1. A new deterministic sampling order.
2. Making that work with a multiscale autoregressive transformer architecture.
3. The somewhat novel notion (to me) that balanced within-scale conditioning lets one use larger scale jumps, but only if the total number of steps in the conditional chain is fixed.

**Additional Comments:**

As someone who doesn't really work in this area, I attempted to read this paper and it took me quite a while to understand. A broader introduction that's more comprehensive might help this paper reach a broader audience.

**Audience:**

Yes

**Audience Explanation:**

To the best of my knowledge, AR image generation seems within the scope of TMLR's audience and this feels like a novel contribution in that space, though one that's only empirically justified.

**Broader Impact Concerns:**

None.

**Claims And Evidence:**

Yes

**Claims Explanation:**

I would argue that the authors show strong empirical evidence through their experiments. The plots on scale-ratio and step tradeoff are particularly convincing. That said, I'm a bit puzzled as to the mechanism. Is there a formal argument for why the total number of steps in the conditional chain is the dominant quantity? Why is this checkboarding the near-optimal one (if it is)? Could the authors make a stronger theoretical argument for why this works?

**Requested Changes:**

I would like to see more theoretical justification, as mentioned above. It would also be useful to have the best results bolded in the tables. It might also be useful to see more than ImageNet latents, but I understand if this is not possible.

---

> ### Author Response · Authors · 2026-04-27
> **discussion responses**
>
> Thank you for your thoughtful review and questions.  We have uploaded a new revision of the manuscript, which includes additional discussion and background material, described below.  We hope these make the paper more accessible and help explain more of its inner workings.
>
> **Mechanisms underpinning the scale-step relationship**
>
> Thank you for this question.  While the findings are at some level necessarily
> empirical from dependence on the data distribution, there are some foundations
> that might help explain the relationship between scales and steps.
>
> Consider the following 4x4 area, with four of the 16 locations labeled for
> reference:
>
> ```
> A . B .
> . . . .
> C . D .
> . . . .
> ```
>
> The question here is, how many locations can we sample independently?
>
> First, consider a 2x scaling ratio.  When scaling up, each pixel at scale i-1
> "turns into" a 2x2 window of pixels in scale i.  In the example above, this
> means that we've already sampled four values in scale i-1, corresponding to the
> 2x2 locations anchored at A, B, C and D.  So for the most part,
> A, B, C and D are conditionally independent of each other, given samples at
> the corresponding locations in the previous scale.
> (In practice, this is not entirely true --- there could be fine-grained textures
> that need to be consistent between A and B in the new scale, for example --- but it's
> mostly the case).  Thus we can sample all four at the same time.
>
> In general, this suggests that when upsampling with a ratio S, so that each
> pixel at scale i-1 "turns into" a window of SxS pixels at scale i, we can
> sample each window in parallel, but the SxS locations within each window may still
> need to be sampled sequentially.
>
> For a S=2 ratio, corresponding to SxS=4, this would imply 4 steps/scale, which
> is what we see in practice.
>
> For a S=4 ratio, corresponding to SxS=16, this would imply 16 steps/scale.
> However, as shown by our measurements in the paper, we can use only 8 and get
> similar results.  That is, A and D in the above grid can be sampled at the same
> time, even though they are within the same SxS window.  Thus, it's possible to
> view the 8 steps/scale as an approximation of the 16 we know are sufficient
> according to the conditional independence induced by upsampling.
>
> A key enabler here is very likely information redundancy: that is, information
> provided by some samples may overlap with the information provided by other
> samples.  In particular, the information provided from samples of A,B,C,D above
> using S=4x upsampling overlaps information that would have been supplied in the
> previous scale using S=2x upsampling.  This is because of the balanced sampling
> order, which spaces out samples uniformly in the image area.  Once A, B, C and
> D are sampled, we are already conditioning on some information from each 2x2
> window, even when scaling up by 4x:  part of the information that would have
> been provided in each 2x2 window is also part of the A, B, C and D sampled
> values.  The redundancy in the information enables more aggressive parallel
> sampling than might be predicted by the sequential-within-each-window
> assumptions --- how much more parallelism is enabled by this mechanism depends
> on the data.
>
> In this paper, we find that we can't go lower than 8 steps/scale for S=4x
> scaling without sacrificing quality.  Moreover, even for values below this
> limit, we find similar degradation profiles for multiple ratios S=2,3,4 when
> measured by total steps.  This leads to the conclusion that the total number of
> steps is a determining factor for a range of scale factors.
>
> We added this discussion to the revision in Appendix C.
>
>
> **Additional background and introduction**
>
> Thank you for the comment.  Upon rereading, we agree that it may progress too quickly
> into details of image autoregressors for many readers.  We added additional background
> material in both the introduction and a new background section (sec 2.1 of the revision).
> We hope this makes our paper more accessible to readers with more varied backgrounds.
>
>
> **Table 1 boldfaced best results**
>
> Thanks for mentioning this.  We've added this formatting to the table.

---

### Review · Reviewer_ftqh · 2026-05-05

**Summary Of Contributions:**

### Summary
This paper introduces a multiscale autoregressive (AR) image generation framework based on a spatially balanced progressive checkerboard sampling order. The key idea is to define a spatially balanced checkerboard ordering that enables parallel sampling at each scale while maintaining balance across all levels of a quadtree subdivision. The paper shows that performance is primarily governed by the total number of sampling steps rather than how steps are distributed across scales. On ImageNet 256x256, the method achieves competitive performance compared to recent AR models while requiring as few as 17 sampling steps.

### Strengths
1. Clear motivation. The paper addresses the trade-off between parallelism and dependency modeling in AR generation.
2. Simple design. The checkerboard ordering is simple, deterministic, and does not require auxiliary tokens.
3. Sampling efficiency. The method achieces comparable performance with significantly fewer sampling steps.
4. Thorough analysis. The paper includes ablations on scale ratios, entropy dynamics, and RoPE mixing.

### Weaknesses
1. The experimental results in Table 1 are not strong. The FID scores of Checkerboard-L are not the best or second-best even among AR-VQ methods.
2. The key claim that the total number of steps is the dominant performance factor is evaluated only with the S model in Section 4.3, and confirmed with limited experiments at 512x512. It is unclear whether this holds for the L model, which is the one used in the benchmark table.
3. The ablation scope is limited. There is no ablation of the checkerboard ordering itself versus, for example, a raster or random within-scale ordering at the same step count.

**Audience:**

Yes

**Audience Explanation:**

The paper addresses a well-posed problem: how to make autoregressive image generation faster without sacrificing quality. The finding that the total number of steps dominates performance independent of scale ratio simplifies a key design decision in multiscale AR models, and is likely to be of direct interest to practitioners building such systems. The entropy analysis and RoPE mixing experiments also contribute to a mechanistic understanding of how these models condition on spatial context, which is of broader interest to the machine learning community.

**Broader Impact Concerns:**

The paper focuses on image generation, which carries the standard risks associated with generative models, including potential misuse for creating synthetic media that could be used for misinformation or deepfakes. The paper does not include a Broader Impact Statement. Given that the generated images are class-conditional on ImageNet categories, the immediate risk profile is relatively modest. Nevertheless, a brief Broader Impact Statement acknowledging these risks would be appropriate.

**Claims And Evidence:**

Yes

**Claims Explanation:**

- Claim 1: The progressive checkerboard ordering enables competitive image generation with few steps.
This is supported by Table 1 and Figure 4. The comparison is fair in that model parameters are matched and inference time is measured on the same hardware.
- Claim 2: The total number of steps is the dominant performance factor across scale ratios of 2, 3, and 4.
It is supported by Figure 5 (S model, 256 resolution) and Figure 9 (S model, 512 resolution). The evidence is convincing within these experimental conditions: the FID curves for ratios 2, 3, and 4 overlap tightly when plotted against total steps.
- Claim 3: The checkerboard ordering reduces within-scale mutual dependence, enabling larger scale-up factors.
In Figures 6-7, entropy drops fastest at the midpoint of each scale, which is consistent with each unsampled location first acquiring four adjacent sampled neighbors.

**Requested Changes:**

1. Validate the "total steps dominate" claim for the L model. Section 4.3 and Figure 5 present this finding only for the S model, but the benchmark comparison (Table 1) uses the L model. It is necessary to verify that the claim holds for the larger model.
2. Ablation of the checkerboard ordering. The paper does not isolate the effect of the checkerboard spatial ordering from simply having more AR steps within a scale. An ablation comparing (a) checkerboard order, (b) raster order, and (c) random order at the same total step count and scale ratio would strengthen the claim that the specific spatial structure of the ordering matters.
3. Clarify the CFG selection procedure. Table 1 reports results at specific CFG values (1.4, 1.5, 1.7). It should be made explicit whether these were chosen by a sweep on a held-out validation set or on the same 50k evaluation set used for FID/IS.
4. Discuss the performance gap relative to ARPG and LPD. ARPG achieves FID 2.30 at 32 steps and LPD achieves FID 2.40 at 20 steps; the proposed method achieves FID 2.72 at 17 steps. While the step count is lower, the FID gap is non-trivial. The paper should discuss why these methods achieve better FID and whether the gap is expected to close with longer training or larger models.

---

> ### Author Response · Authors · 2026-05-18
> **discussion response**
>
> Thank you for your careful reading of our paper and thorough comments.  Your
> suggestions of additional ablations and evaluations in particular further
> strengthened our work.  We have responded to your questions below.
>
>
> **Total steps evaluations for L model**
>
> Thank you for asking about evaluations on the L model to test performance at
> total step counts between scale ratios.  We have included these evaluations
> using the L model in Appendix B of the new revision.  The results continue to
> support the relationship between scales and steps, with FID by total steps
> showing similar curves between 2x and 4x L models.  As in the 512 resolution S
> model experiments, the 2x ratio is slightly better than the 4x ratio, but the
> difference is much less than the effect of step count:  Total steps remains the
> main factor in this setting as well.
>
>
> **Ablation of the checkerboard ordering**
>
> Thank you for this question.  To verify the checkerboard order, we trained a
> new S model using random within-scale ordering, otherwise identical to the S
> model used in our other ablations.  The results are included in Figure 4c of
> the new revision.  Random ordering results in 6.01 FID, while checkerboard
> achieves 5.32, an improvement due to its spatial balancing.
>
>
> **Clarify the CFG selection procedure**
>
> We use the same procedure as other methods including PAR and RandAR, so our
> results are directly comparable.  All methods scan at 0.1 increments using the
> reference set, and we follow the same procedure.  We have clarified this
> in the revision.
>
>
> **"The experimental results in Table 1 are not strong"**
>
> We respectfully disagree, and believe our results are indeed strong:  our
> method is among the best performing, competitive with ARPG and LPD on the
> combined Pareto front of FID, IS and Step Count, even as it uses simpler
> sampling procedures.
>
>
> **Discuss the performance gap relative to ARPG and LPD**
>
> As we point out above, the overall picture for performance is competitive with
> these methods.  However, we acknowledge that the minimum FID is larger.  The
> reason for this remains unclear.  The training schedule is probably one of the
> largest differences that might have affected this, but we discuss a few
> possibilities below.
>
> 1. *Training Schedule*: We use a batch size of 64 for 200 epochs, increased to
> 320 for the last 5 epochs with gradient accumulation.  LPD and ARPG use batch
> sizes 2048 and 1024, respectively, for over 400 epochs.  Thus, our training
> schedule uses fewer total examples and has noisier gradients.  Importantly, our
> 5x batch size increase at the end resulted in a performance improvement, and
> increasing further to 2048 would be an additional 6x increase, likely leading
> to further improvements.  This may cover all or most of the difference.
>
> 2. *CFG Schedule*: All methods are highly sensitive to the CFG schedule, with
> CFG values searched up to 0.1.  However, ARPG uses a linear CFG schedule with
> arccos block progression, similar to MAR, while we use piecewise-constant CFG,
> similar to PAR.  Experimenting further with linear schedules, we saw no
> substantial difference in minimum FID.  So this doesn't appear to be a
> substantial difference.
>
> 3. *Sample vs Output Attention Split*:  ARPG uses different phases for
> self-attention in sampled conditioning inputs, followed by exclusive
> cross-attention between these features and the output query positions being
> sampled.  LPD uses an interleaved training mask to produce a similar split
> between attention for samples and queries, where queries encode positions
> alone.  Our method processes samples and queries simultaneously using an input
> mixture, with no explicit split, and uses the upsampled previous-scale samples
> as part of the queries in addition to position.  Thus, while the queries may
> include richer information useful for generation, we also don't discard their
> processing from the KV cache the way LPD does.  This results in half as many
> sequence entries during training, but at the potential cost of retaining some
> of the KV vector capacity for query processing that may not be as useful for
> later steps.  While we did not explore the impact of this, including such a
> split may be a promising way to combine all three methods' approaches.
>
>
> **Broader Impact Concerns**
>
> Thank you for this suggestion.  We have added a Broader Impacts section
> to the new revision.

---

### Decision · Action_Editor_2Ran · 2026-06-08

**Recommendation:** Accept with minor revision

**Additional Comments:**

The reviewers are generally satisfied with the authors' responses. However, a few minor issues could still be addressed to further strengthen the paper. The "total steps dominate" conclusion is currently supported only by ImageNet experiments with a single tokenizer, and thus its generality is unclear. The entropy analysis needs more formal information-theoretic or statistical evidence. The 512×512 experiment is based on transfer rather than training from scratch and therefore provides only limited additional validation. While the method is compared with ARPG and LPD, it is better to have a more detailed discussion of the reasons behind the observed performance differences.

**Audience:**

Yes

**Audience Explanation:**

All reviewers believe researchers in autoregressive image generation and efficient multiscale sampling would find the findings relevant and interesting.

**Claims And Evidence:**

Yes

**Claims Explanation:**

All reviewers agree that the experimental evidence is considered accurate and convincing.

---

> ### Author Response · Authors · 2026-07-03
> **camera-ready version**
>
> Thanks for your reading and review.
>
> We have uploaded the camera-ready version with requested revisions including our discussion on ARPG and LPD comparisons, as well as statistical significance checks on the entropy analysis.  These are in a new paragraph in sec 4.2, and new appendix D, respectively.  We believe the points on 512 resolution transfer limitations and reliance on imagenet with single tokenizer are already appropriately acknowledged and discussed, but have made a few small edits to make this clearer.